# Postharvest Ultraviolet Radiation in Fruit and Vegetables: Applications and Factors Modulating Its Efficacy on Bioactive Compounds and Microbial Growth

**DOI:** 10.3390/foods11050653

**Published:** 2022-02-23

**Authors:** Magalí Darré, Ariel Roberto Vicente, Luis Cisneros-Zevallos, Francisco Artés-Hernández

**Affiliations:** 1LIPA—Laboratorio de Investigación en Productos Agroindustriales, Universidad Nacional de La Plata, Calle 60 y 119 s/n, La Plata CP 1900, Argentina; lipa@agro.unlp.edu.ar; 2Department of Horticultural Sciences, Texas A&M University, College Station, TX 77843, USA; lcisnero@tamu.edu; 3Postharvest and Refrigeration Group, Department of Agronomical Engineering & Institute of Plant Biotechnology, Universidad Politécnica de Cartagena, 30203 Murcia, Spain; fr.artes-hdez@upct.es

**Keywords:** UV, UVB, UVC, UV illumination, photochemical treatments, abiotic stress, antioxidants, phytochemicals, quality, food safety

## Abstract

Ultraviolet (UV) radiation has been considered a deleterious agent that living organisms must avoid. However, many of the acclimation changes elicited by UV induce a wide range of positive effects in plant physiology through the elicitation of secondary antioxidant metabolites and natural defenses. Therefore, this fact has changed the original UV conception as a germicide and potentially damaging agent, leading to the concept that it is worthy of application in harvested commodities to take advantage of its beneficial responses. Four decades have already passed since postharvest UV radiation applications began to be studied. During this time, UV treatments have been successfully evaluated for different purposes, including the selection of raw materials, the control of postharvest diseases and human pathogens, the elicitation of nutraceutical compounds, the modulation of ripening and senescence, and the induction of cross-stress tolerance. Besides the microbicide use of UV radiation, the effect that has received most attention is the elicitation of bioactive compounds as a defense mechanism. UV treatments have been shown to induce the accumulation of phytochemicals, including ascorbic acid, carotenoids, glucosinolates, and, more frequently, phenolic compounds. The nature and extent of this elicitation have been reported to depend on several factors, including the product type, maturity, cultivar, UV spectral region, dose, intensity, and radiation exposure pattern. Even though in recent years we have greatly increased our understanding of UV technology, some major issues still need to be addressed. These include defining the operational conditions to maximize UV radiation efficacy, reducing treatment times, and ensuring even radiation exposure, especially under realistic processing conditions. This will make UV treatments move beyond their status as an emerging technology and boost their adoption by industry.

## 1. Introduction

Ultraviolet (UV) radiation comprises the region of the electromagnetic spectrum (EM) between visible light and X-rays (100–400 nm) [1]. It was discovered in 1801 by Johann Wilhelm Ritter, who observed that radiation outside the violet end of the visible solar spectrum could decompose silver chloride [2]. Seven decades later, it was discovered that UV light could prevent microbial growth [3].

The UV region is frequently divided into three sub-regions, UVA (315–400 nm), UVB (280–315 nm), and UVC (100—280 nm) (Figure 1), which are used for CIE and ISO standards [4]. Further sub categorization has been performed by some authors to discriminate within the UVC region vacuum UV (100 and 200 nm) with stronger ionizing power, but less penetration [5,6,7]. Based on the known mechanisms of plant photoreception, the UVA region has been split into short-UVA (315–350 nm) and long-UVA (350–400) [8].

## 2. UV Radiation Sources

Commercially available UV sources include mercury lamps (low- and medium-pressure), pulsed light (PL), and light-emitting diodes (LEDs) [9]. Although the market has become highly dynamic with the improvement of LEDs, it is still dominated by mercury lamps. These sources are based on the excitation of gas discharges and feature several pitfalls, including a relatively high voltage requirement to operate and a substantial amount of heat released [10]. However, one advantage is that, especially with medium-pressure Hg lamps, high output powers can be achieved.

Xenon inert-gas lamps were introduced during the late 1970s in Japan, leading to the development of sterilizing technology, called PureBright^®^ [11]. PL treatments consist in the exposure of fresh produce to polychromatic light (200–1100 nm), including ultraviolet (180–400 nm), visible (400–700 nm), and near-infrared (700–1100 nm) wavelengths, in the form of intense, but short, pulses (1 μs–0.1 s) [12,13].

Light-emitting diodes (LEDs) are based on the junction of two-terminal semiconductors (p-n junction) converting electricity into radiation. Depending on the materials out of which the semiconductors are made, the LEDs emit at different wavelengths [14]. The first LEDs, in the early 1960s, emitted infrared (IR) light. Over the years, it became possible to develop LEDs of shorter wavelengths. UV LEDs have several advantages relative to mercury lamps, including their lack of requirement for warming time, their lack of mercury, their compactness, their robustness (with UV LEDs, no protection against glass breakage is necessary, and mobile use is possible), and their large wavelength diversity (210 nm to 360 nm by varying the aluminum content in the AlGaN quantum wells) [10]. In addition, they have lower electromagnetic interference, are easily adaptable for fast modulation in terms of radiation intensity and pulse duration, present narrow-band emission without spurious peaks, and require low maintenance [15]. Two important advantages of LEDs are their long lifespan (expected lifetimes of many 10,000 s of h) and low heat emission [16]. The Achilles heel of UV LEDs is their relatively low quantum efficiency [17]. However, in recent years, by reducing dislocations and defects and improving semiconductor doping and light extraction, their quantum efficiency has been increased [18].

## 3. Uses in the Food Industry

UV technology has been applied in the food industry for many different purposes (Figure 2).

Surface sterilization: One of the most common uses of germicide UVC lamps is as environmental sterilizers in foodstuffs filling equipment, conveyor belts, containers, and working surfaces [19,20]. Sterilizing UV lamps are frequently used for aseptic packing, a technology that is expected to continue growing in the coming years [21,22].

Fluid disinfection: UV radiation in the C zone has been used for water disinfection since 1909. It has been also applied for juice pasteurization [23]. UVC does not generate undesirable by-products, but on the other hand, it does not provide residual disinfection capacity [24]. It has been applied to reduce chlorine use.

Air treatment: Air disinfection can be achieved through different strategies, ranging from irradiating just the air in the upper region to treating all air, either when the room is empty or during circulation through air-conditioning systems [25]. The fact that UVC relatively low radiation doses (0.1–0.3 kJ m^−2^ for 2 log cycle reductions) can inactivate human SARS-coronaviruses has increased the recent interest in using UV radiation for air treatment [26,27,28].

Waste treatment: Another application of UV radiation has been the elimination of undesirable volatile organic compounds (VOCs) in industrial exhausts [29,30]. This has been achieved through advanced oxidation processes combining UV radiation with photocatalysts, such as TiO_2_ [9,31]. This strategy generates highly oxidative environments, which facilitates the degradation of unwanted molecules [32].

Insect trapping: For a long time, it has been known that UV radiation can attract insects; thus, it is used for trapping purposes [33]. The most common insect light traps use “black-light” fluorescent tubes emitting ultraviolet (UVA) as an insect attractant in both pre and postharvest [34]. Furthermore, insects may be trapped in glued materials or killed in electrically charged grids [35].

## 4. Uses in Fruits and Vegetables Postharvest

UV technology may be of interest for the postharvest treatment of fruits and vegetables for many different purposes [36,37] (Figure 3).

Raw material selection: The presence of skin defects or wounding is a main factor affecting consumer acceptability and purchase decisions. Consequently, one of the most intensive activities of packinghouses is to separate fruit with these defects. Normally, this is performed through visual inspection or machine optical sorting when the fruit is illuminated under proper white light [38]. In citrus, the use of UV lamps during initial classification has may facilitate the identification of physical damages. UVA “black light” illuminates fruit, showing that small peel cracks fluoresce intensely, allowing segregation at early classification steps [39]. The conveyor belts transporting the fruit cross these rooms, where the operators must wear protective glasses and gloves.

Control of spoilage and pathogenic microorganisms: Relying on UVC germicide properties, a large body of research evaluated this technology in fruits and vegetables to control surface microorganisms [40]. Microbial death induced by UVC has been attributed to DNA mutations, including the formation of cyclobutyl-type dimers (pyrimidine dimers) and pyrimidine adducts [41,42]. Furthermore, the overproduction of reactive oxygen species (ROS) induced by UV radiation can oxidize membrane lipids and inhibit critical cellular enzymes [43]. Most enzymes that contain aromatic amino acids are likely to be sensitive to UV radiation to some extent due to their absorption in this region. Due to its higher energy, UVC is the most effective at killing microorganisms [44]. UV radiation is lethal to bacteria, viruses, protozoa, and fungi [45]. Successive studies showed that UV radiation is more efficient for inactivating Gram-negative than Gram-positive bacteria. This effect has been associated with the difference in the cell wall peptidoglycan structure, which can affect radiation penetration [46]. Furthermore, eukaryotic organisms are normally more resistant to UV than bacteria due to their higher cell size, complexity, and genetic redundancy [47]. The relatively high yeast resistance to UV radiation has also been associated with lower DNA pyrimidine content relative to bacteria, which may increase the likelihood of photons being absorbed by other compounds [48].

Several works have studied the impact of postharvest pre-storage single exposure on most common postharvest fungal pathogens, including *Rhizopus*, *Penicillum digitatum*, *P. expansum* and *Penicillum italicum*, *Monilinia* sp., *Botrytis cinerea*, *Colletrichcum* sp. and *Fusarium* sp. among others [49,50], with positive effects with regards to reducing disease incidence and severity. Direct germicide action compromising microbial viability has been frequently reported [51], but less severe effects, such as the reduced germination speed of viable conidia, have also been observed [52]. With regards to human pathogens, the direct UV irradiation of fresh produce reduced the viability of *E. coli*, *Salmonella*, and *Listeria* [53,54,55,56,57]. These studies have so far mostly been conducted on a laboratory or, in the best scenario, pilot scale. A review of the available research suggests that their widespread commercial use requires some important aspects to be solved, especially with regards to the adaptability of this treatment to continuous processing lines (where the treatment duration may require minutes at low irradiances), safety procedures for workers and even radiation exposure, while avoiding mechanical damage, especially to fruits and vegetables. Furthermore, the fact that wet cleaning methods have long been applied to fresh produce could limit the fast adoption of a different technology.

The mechanisms through which UV prevents deterioration exceed radiation germicide properties and involve host-induced physiological responses [58,59,60]. They include the inhibition of ripening-related genes [61] and the induction of enzymes and compounds, improving tolerance to opportunistic pathogens or other environmental effectors causing oxidative damage. An array of defensive responses has been observed in UV-stressed tissues. It includes the activation of glucanases and chitinases thought to be involved in the degradation of microbial cell walls, the induction of genes related to phenolic compound biosynthesis or oxidation, such as phenylalanine ammonia lyase (PAL), polyphenol oxidases (PPOs), and peroxidases (PODs) [62,63]. Another frequent change reported is the upregulation of gene coding for antioxidative enzymes, such as superoxide dismutases (SODs), ascorbate peroxidases (APXs), and catalases (CATs) [64]. More recently, Rabelo et al. [65] proposed that the UV response is mediated by the generation of oxidative stress as the primary signaling molecule generated through the partial ionization of water and an increase in mitochondrial activity. With regards to the induction of compounds that could contribute to increasing host tolerance to pathogen attack, there have been several reports on metabolites with direct antimicrobial activity (i.e., hydroxycinnamic acid derivatives, 6-methoxymellein, scoparone, scopoletin, rishitin), or reinforcing structural barriers [49,66]. Phenolic compounds showing in many cases antimicrobial and antioxidant properties have been the most frequently identified family of induced antimicrobial compounds.

Enhancement of bioactive compounds: Early on, in 1977, Langcake and Pryce [67] showed that UV exposure induced bioactive compounds in grapes. This raised the interest in using UV treatments to improve fruit and vegetable phytochemical profiles. A literature review shows that the number of publications on bioactive compounds and UV radiation has increased exponentially in the last three decades. Table 1 provides an overview of some relevant studies that used UV radiation on whole and fresh-cut fruit and vegetables, focusing on bioactive compounds. As a rule, the results show that the induction of antioxidants by UV radiation tends to be greatest for phenolic compounds, although increases in other antioxidant metabolites (ascorbic acid, glutathione, carotenoids) have also been reported [65,68]. The subclass of compounds induced is mostly dependent on the species considered [69,70] and even on the cultivar [71]. Increases in phenolic acids, non-anthocyanin flavonoids, anthocyanins, other flavonoids, isoflavones, and stilbenes have been frequently reported [62,72,73,74,75,76]. Furthermore, the subregion of the UV spectra applied may determine the type of metabolite elicited; in carrots, chlorogenic acid and isocoumarin were more inducible by UVB and UVC radiation, whereas ferulic acid was elicited by all UV regions to comparable levels [77,78]. At low irradiances, UVB and UVC regions are considered more inductive of secondary metabolites than UVA and UVB, which still have effects on antioxidants but also initiate several photomorphogenic responses [79,80]. Other factors affecting phytochemical accumulation besides the UV region and dose that have been less studied include the irradiances of the illuminating source and the mode of exposure (pulse number duration, interval between successive irradiations, etc.) [13,81,82]. Recently, UV has been reported as a suitable green strategy to enhance the nutraceutical content of fruit and vegetable beverages [83]. Considering there are receptors for UVB and UVA but no receptors for UVC, it has been proposed that ROS plays a key role in secondary metabolite biosynthesis for all three UV lights [77,78]. However, further studies are needed to confirm the role of oxidative stress through UVB and UVA receptor-mediated responses. Another proposed hypothesis is that UVB and UVC could share the same photoreceptors, since the action spectrum of UVR8 protein (as the main UVB receptor) ranges from 250 to 310 nm [8], including the UVC region; therefore, this UVB receptor could also be activated by UVC. It can explain why UVC produces similar effects to UVB in some cases according to previous research, since the energy provided by shorter wavelengths of UVC may activate the UVR8 protein [84]. Nevertheless, the UVR8 spectrum of action could slightly vary, depending on the metabolic pathway. The action spectrum on PAL induction is closer to the UVB region and thus may explain why UVB or even the combined effect of UVC + B obtained a higher increase in total phenolics than UVC [85].

In addition to their effects on phenolics, carotenoids and ascorbic acid, UV treatments have been shown to increase vitamin D content. Exposure to sunlight and dietary foods are the most important ways for humans to obtain vitamin D [118,119,120]. Lifestyle changes due to the SARS-CoV-2 pandemic have substantially reduced our regular exposure to the sun, resulting in vitamin D deficiency [120]. Furthermore, patients with vitamin D deficiency were five times more likely to be positive for COVID-19 than patients with no deficiency [121]. Mushrooms are rich in ergosterol, a precursor to vitamin D2, which can be converted to vitamin D2 under proper UV exposure. The eliciting capacity of UVA, UVB, and UVC has been tested in different types of edible mushrooms, increases ranging between 25 and 8000% reported [122]. The UVB zone of radiation shows the greatest inductive effect, with other important factors being the radiation dose applied, the product’s water content, and the degree of processing [123].

Retardation of ripening and senescence: In some commodities and under proper treatment conditions, UV radiation may delay ripening and senescence [60,124]. These effects could be understood by recognizing that both developmental processes are genetically regulated and require specific transcriptional programs to be induced. Under UV radiation, cells redirect their normal developmental programs to primarily respond to external stressing stimuli [125,126]. Stress acclimation favors the induction of UV-responsive genes at the expense of ripening and or senescence-related genes [61,127,128]. The efficacy of UV treatments to retard ripening is certainly dependent on the UV irradiation schedule used, and to a great extent on the initial ripening state of the commodity [129]. Ripening progression would be more effectively delayed in fruit treated at early stages. Hundreds of genes are up- and down-regulated in fruit irradiated with UV radiation. Most frequently, up-regulated genes are mainly involved in signal transduction, defense response, and metabolism. Conversely, genes related to cell wall disassembly, photosynthesis, and lipid metabolism are usually suppressed. The retardation of ripening-related changes such as softening may be one important contributor to the improved tolerance to postharvest spoilage pathogens observed in treated products [61,82,130]. In some cases, especially with long-term exposure, UVB has been shown to promote fruit ripening [131].

Induction of cross-stress resistance and synergistic responses: It is currently known that biotic and abiotic stress responses use common signals, pathways, and triggers [40]. This overlap includes common changes in cellular redox status, reactive oxygen species, hormones, protein kinase cascades, and calcium gradients as common elements [131] and helps to explain cross-tolerance phenomena, whereby exposure to one type of stress can improve tolerance to several different types of stress [132]. UV treatments preceding cold storage have been reported to improve the chilling tolerance of sensitive commodities such as peach [133], sweet pepper [134], and tomato [135]. Some of the metabolic changes behind cross-tolerance include the induction of polyamine biosynthesis in stone fruit and an increase in antioxidant enzymes in the case of pepper [64]. Another effect observed from combining stresses is a synergistic response in plant tissues, as in those reported with wounding and UV exposure. This synergism among stresses applied simultaneously is due to the activation of similar signaling molecules and signaling pathways [136]. This has been reported in the biosynthesis of polyphenols [77,78,102] and betalains [137]. In Table 2, the physiological responses could potentially be higher in fresh-cut products compared to whole tissues, where synergistic effects take place due to skin removal in whole produce, being skin-determinant in the response due to the partial blockage of UV penetration [77].

Advantages and disadvantages of UV treatments: UV technology provides several advantages over other conventional preservation methods, but also has some important drawbacks (Table 2). UV treatments can be simply applied and are able to inactivate a wide range of pathogenic and spoilage microorganisms while causing negligible changes in nutritional and sensory quality [95,96]. In contrast to other disinfection practices, it does not require water or generate wastes and leaves no residues on treated surfaces and foods [48]. In addition, it is approved with no major restrictions by the European Food Safety Authority (EFSA), the US Food and Drug Administration (FDA), and most other food regulatory agencies. When compared to electron beam or gamma irradiation, UV technology also offers several advantages (i.e., lower investment required, fewer regulations, no treatment label needed in the products) [138]. Another plus for UV treatments relates to their ability to be relatively simply combined with other preservation techniques in the search for additive or even synergistic effects [70,139]. UV irradiation is considered a very valuable tool within the hurdle technology, an integrated approach aimed in creating safe and stable foods by combining multiple physical, chemical, and/or biological preservation methods [139]. UV treatments have been applied in metabolically active fruit and vegetables not just due to their germicide action but to activate natural defense mechanisms (i.e., phytoalexins, free radical scavengers’ antimicrobial and antioxidant enzymes) [140]. Compared to many conventional methods, UV treatments are also energy-efficient and cost-effective [141]. Finally, UV has historically being applied for a wide range of applications and validation data is available.

However, there are several limitations of UV technology which prevent its wide use in the food industry. It has a low penetration power, especially in solids and turbid media [142]. Moreover, the fact that it does not provide residuality may be a drawback when long-lasting effects are desired. To properly express the UV germicide properties direct and even exposure is required [143]. This may be challenging in some cases under commercial operations. Another difficulty could be to adapt UV treatments to continue processing lines, especially if long exposure times or low irradiances are required. Considering the moments in which UV could be used, UV treatments are defeated compared to ionizing irradiation, since they cannot be applied after packing [144]. UV radiation could be also harmful to operators if direct exposure occurs, but proper safety procedures and personal protective equipment can easily prevent such risk.

Factors determining UV treatment efficacy: In the last two decades, more than 500 publications have tested postharvest UV illumination strategies in fresh horticultural produce. These studies have been useful in identifying the main factors determining the efficacy of the technology on both the commodity and the treatment sides (Table 3).

The factor that has received the most attention is the type of commodity. So far, most commercially relevant fresh fruit and vegetable (>100 products) have been tested. In general, positive results have been found in one or more of the effects outlined in the previous section, depending on the species considered. In a few cases, some damages, mostly related to tissue discoloration, have also been reported [145,146]. The outcome of UV treatments has also been reported to be dependent on the cultivar [71,147]. A third relevant factor with regards to UV irradiation efficacy is the product-ripening stage. As a rule, the treatment of unripe fruit may lead to stronger phenotypes regarding ripening delay than fruit at advanced maturity stages. The ripening stage not only affects the impact of UV irradiation on ripening changes, but also its ability to control postharvest decay. In fresh-cut bell peppers, UV radiation was more effective at reducing soft rot in red ripe fruit [148,149]. The degree of commodity processing also has a substantial impact on quality maintenance and antioxidant elicitation [102,114]. This could be due to the increased surface during processing, which results in the exposure of a greater area of the product. However, the induction of synergic effects induced by simultaneous UV and wounding stresses has been suggested to be involved as well [70,100]. Finally, the nature of the interphase between UV radiation and the product can also have a relevant effect. For instance, surface wetting would be expected to reduce UV penetration in vegetable tissues. Another effect may be the impact of interphases on microorganism arrangements. Heterogeneous microorganism distributions in liquid droplets can lead to preferential concentration in the outer layer at the liquid–air interface, which may protect the cells inside the droplet from UV bactericidal action [55].

Regarding treatment, the subregion of the UV spectra employed has a large influence on the kind of responses obtained [150]. For instance, changes in nutraceutical compounds have been reported for all three zones, but the greatest impact on disease control is clearly observed for UVC [151]. The second relevant process variable is the radiation dose (energy per unit area). A broad range of doses has been explored (i.e., 0.1–50 kJ m^−2^ for UVC region). However, a single study tried more than two or three doses and complete optimization studies are currently lacking. Working with strawberries, Cote et al. [152] showed that the radiation intensity (energy per unit time and unit area) is another key process factor determining the efficacy of treatments. Strikingly, this variable has been overlooked in several studies that do not report the radiation intensity of the illuminant used. Subsequent work by Darré et al. [81] reported that both radiation dose and intensity should be considered simultaneously when optimizing UV treatments. Low UVB doses (5 kJ m^−2^) and intensities delay chlorophyll degradation and may be useful to complement refrigeration. Instead, high-intensity UVB exposure may be better suited for the freezing industry as a pre-treatment to increase the antioxidant capacity prior to further processing.

The irradiation pattern is another aspect to be considered when selecting a proper treatment schedule. This has been relatively well studied for pulsed treatments with xenon lamps. Bauer et al. [153] found that UV germicidal efficacy against *Bacillus* spores was a function of pulsed radiation parameter, with shorter pulses and lower frequencies being more effective. With regards to treatments with conventional single-UV radiation sources, almost no attention has been paid to the relevance of the radiation exposure pattern. Ortiz-Araque et al. [82] showed that at the same total dose (4 kJ m^−2^) and intensity (36 W m^−2^), fractionated (two-step and five-step) treatments were much more effective at controlling softening and decay than single pre-storage irradiation. The fractionation of the treatments during storage delayed pectin debranching and delayed the solubilization of polyuronides [130].

Although treatment uniformity and ensuring that all food surfaces are exposed to UVC light may be one of most problematic causes in commercial settings, there are very few studies attempting to overcome this limitation. Finally, some studies have reported that light exposure after treatment may affect the outcome of UV treatments. This may be due to photo repair mechanisms that may be activated in the presence of light [11,154,155]. It is not clear whether this effect would be significant in products. It is important to state that eventually, all these factors and their combinations result in the generation of signal molecules, such as ROS, and ROS levels determine the molecular and physiological response of stress-challenged produce, either staying in homeostasis, going through hormesis, or even responding to extreme stress [136].

## 5. Concluding Remarks

After microbial control, the elicitation of bioactive compounds is the aspect that has received the most attention with regards to the use of postharvest UV treatments in fresh produce. So far, several works have reported that appropriate exposure to UV radiation may stimulate the biosynthesis of phenolic compounds and, to a lesser extent, of ascorbic acid, carotenoids, and/or glucosinolates. Recent studies have identified the most relevant factors determining the nature and extent of such changes on both the commodity and process sides. These include the species, cultivar, ripening stage, degree of processing, radiation-product interphase, radiation wavelength, dose, intensity, exposure pattern (i.e., pulse frequency, duration), and illumination regime after treatment.

Despite the significant advances achieved, some important limitations remain. These has likely slowed down the adoption of UV technology by industrial stakeholders. For fresh products, the main challenges remaining are reducing the treatment times to facilitate their compatibility with continuous processing lines and increasing the treatment uniformity for large produce volumes without causing mechanical damage to products. Solving these challenges would likely help UV irradiation to move beyond the stage of emerging technology and translate all the knowledge accumulated about its application into production.

## Figures and Tables

**Figure 1 foods-11-00653-f001:**
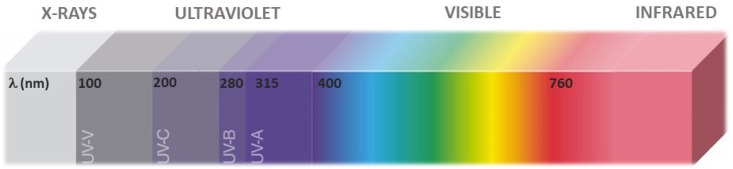
Subregions of the UV spectrum relevant for technological applications and plant photoreception.

**Figure 2 foods-11-00653-f002:**
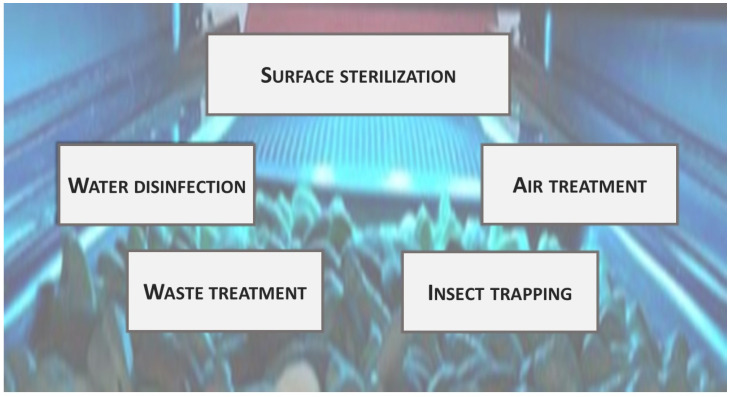
UV radiation applications in the food industry.

**Figure 3 foods-11-00653-f003:**
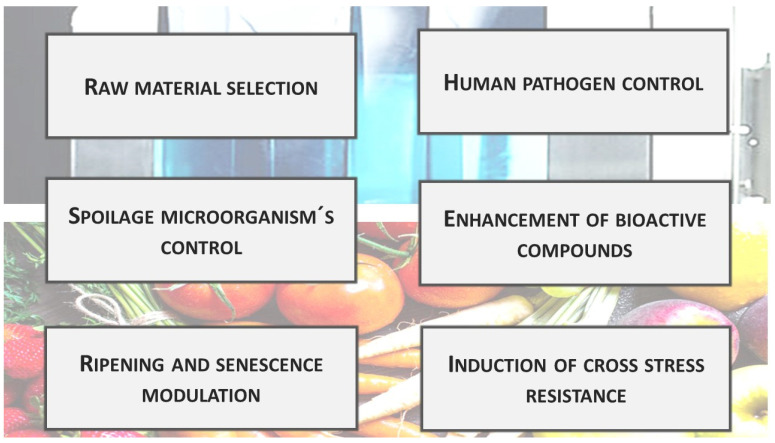
Main uses of postharvest UV treatments for direct application in fruit and vegetables.

**Table 1 foods-11-00653-t001:** Effects of UV radiation on the phytochemical profile of fruit and vegetables and on other quality attributes.

UV Region	Product	Treatment Conditions	Main Results Found	Reference
**UVA**	Blueberries	Dose: 6 kJ m^−2^	UVA had lower inductive effect than UVB or UVC	[86]
	Fresh-cut carrot	Intensity 12.73 W m^−2^ Dose: 45.8 kJ m^−2^	Induction of PAL and increase in total antioxidant capacity and phenolics.	[77]
	Lettuce	Intensity: 3.7 W UVA, 4.2 W UVB, 7.5 W UVC.	Induction of PAL and phenolic antioxidants in all UV regions. UVA caused no growth inhibition. UVB inhibited growth and UVC caused most severe lesions.	[87]
	Tomato	λ: 353, 365 or 400 nm. Intensity:0.28–0.33 W m^−2^ Dose: 0.17–7.1 kJ m^−2^	All wavelengths increased phenolics and carotenoids.	[88]
		Daily dose: UVA 2.9 kJ m^−2^ UVA 11.5 kJ m^−2^ UVB 0.941 kJ m^−2^ UVB 0.353 kJ m^−2^	Increased antioxidant capacity and flavonoid accumulation. UVA was more promising than UVB with regards to firmness maintenances and antioxidant elicitation.	[89]
	Soybean sprouts	UVA 173.0 kJ m^−2^ UVA 346.0 kJm^−2^ Intensity: 2 W·m^−2^	Treatments elicited isoflavone and flavonol accumulation.	[90]
**UVB**	Apples	Dose: 219 kJ m^−2^	Increased content of flavonoids (64%) and hydroxycinnamic acids (38%) in the peel after 14 days.	[91]
	Bell pepper	Dose: 9.0 kJ m^−2^. Storage 4 d at 20 °C under retail sale photoperiod (14 h fluorescent + 10 h Blue & Red LEDs)	Capsaicinoids increased by ~22%, ~38%, and ~27% in the content of capsanthin, capsanthin laurate, and capsanthin esters, respectively, after the UVB treatment. This effect was enhanced by ~18% after an LED-supplemented photoperiod.	[92]
	*Brassicaceae* sprouts	Dose: 15.0 kJ m^−2^	Increased the total phenolics and antioxidant activity. Increased the glucosinolate content by ∼30%. Sulforaphane was enhanced by 37.5% in broccoli sprouts. Sulforaphene was increased by 72% in radish sprouts.	[93]
	Broccoli	Intensity: 3.2–5.0 W m^−2^ Dose: 2–12 kJ m^−2^	Low doses and intensities delayed chlorophyll degradation, whereas high intensity elicited antioxidant accumulation.	[81]
		Dose: 1.5–7.2 kJ m^−2^	UVB increased glucobrassicins by 18–22%. Glucoraphanin was enhanced by 11% in florets exposed to 1.5 kJ m^−2^, while a dose of 7.2 kJ m^−2^ by 16%. Florets exposed to 1.5 and 7.2 kJ m^−2^ UVB increased hydroxyl-cinnamic acids by 12%.	[94]
		Dose: 5–15 kJ m^−2^ alone or in combination with UVC (9 kJ m^−2^).	Combination of moderate UVB and UVC doses reported the highest inductive effect on phenolics and total antioxidant activity. A high UVB dose (15 kJ m^−2^), single or combined with moderate UVC, induced a higher level of glucoraphanin and sulforaphane.	[95]
	Fresh-cut carrot	Dose: 1.5 kJ m^−2^ alone or in combination with 4.0 kJ m^−2^ UVC	UVB caused the largest increase in phenolics and antioxidant accumulation after 3 days at 15 °C.	[77]
		Intensity: 12.73 W m^−2^ (UVA) 10.44 W m^−2^ (UVB) 11.8 W m^−2^ (UVC) Dose: 46–275 kJ m^−2^ (UVA) 37.5–225.5 kJ m^−2^ (UVB) 42.5–255 kJ m^−2^ (UVC)	Phenolics (chlorogenic acid and its isomers, ferulic acid, and isocoumarin), antioxidant capacity, and PAL activity increments. Chlorogenic acid was induced by all UV radiations but mostly by UVB and UVC.	[77]
	Kale sprouts	Dose: 0, 5, 10, and 15 kJ m^−2^	Enhanced the total antioxidant activity by 20%. Doses of 10 and 15 kJ m^−2^ stimulated the glucoraphanin and glucobrassicin synthesis by 30%.	[96]
	Lemon	Dose: 22 kJ m^−2^	Increased levels of anthocyanins, flavonols and flavanones-dihydroflavonols. Increased antifungal activity of flavedo extracts against *Penicillium digitatum.*	[97]
	Mango	Dose: 5 kJ m^−2^	Increased ascorbic acid (42%) and phenolic compound (36%).	[98]
	Red cabbage sprouts	Dose 10 kJ m^−2^ proportionally applied on germination days 3, 5, 7, and 10 days,	Phenolics were increased by 40%, while total antioxidant activity and flavonols content was increased by 35 and 30%, respectively. Carotenoids were also enhanced.	[99]
	Peach and nectarine	Dose: 73–219 kJ m^−2^	Cultivar-dependent response: the stimulation of phenol accumulation occurred after 24 h in ‘Big Top’ (69%) and 36 h in ‘Suncrest’ (21%). Decreased phenolics in of ‘Babygold 7′ after 36 h.	[100]
		Dose: 1.39 and 8.33 kJ m^–2^	Transient increase 24 h after illumination, especially for flavanols, flavonols, and flavones (+123, +70, +55, and +50%, respectively). Phenolics induced not only in the peel but also in the pulp. UVB increased the glycoside/aglycone ratio of flavonols and anthocyanins.	[74]
	Prickly pear (red)	Intensity: 6.4 W·m^−2^ Dose: 5.76 kJ m^−2^	Highest phenolic accumulation. The main phenolics were quercetin, sinapic acid, kaempferol, rosmarinic acid, and sinapyl malate, showing increases of 709.8%, 570.2%, 442.8%, 439.9%, and 186.2%, respectively.	[101]
		Intensity 6.4 W m^−2^ Dose: 5.76–69 kJ m^−2^	Immediate accumulation of betalains (33–40%) and ascorbic acid (54–58%) in the pulp and peel of wounded tissue.	[70]
**UVC**	Blueberry	Dose: 4.0 kJ m^−2^	Increased anthocyanins (70%). Antioxidant enzymes induced (SOD, APX).	[102]
	Broccoli	Intensity: UVB s of 9.27 and UVC 25.21 W m^−2^, Dose 5, 10 or 15 kJ m^−2^ UVB, UVC: 9 kJ m^−2^.	UVB + UVC increased glucobrassicin (34%) at 15 °C. UVB15 + C induced the highest glucoraphanin levels of florets after 72 h at 15 °C. UVB10 + C induced the highest total phenolic content increase (110%) in leaves.	[95]
	Carambola	Dose: 13 kJ m^−2^	UVC induced antioxidant enzymes (CAT, POX and SOD) and phenols accumulation.	[103]
	Carrot	Dose: 9 kJ m^−2^ + hyperoxia (80 kPa O_2_)	Increase in phenolic compounds, which was also observed in hyperoxia for 72 h. UVC + hyperoxia showed higher accumulation of chlorogenic acid.	[104]
	Fresh-cut watermelon	Dose: 1.6–7.2 kJ m^−2^	Increase in antioxidant capacity (7%), maintenance of lycopene and ascorbic acid. Microbial growth retardation. Only the lowest doses (1.6 and 2.8 kJ m^−2^) preserved sensory attributes.	[105]
	Fresh-cut Bimi^®^ Broccoli	Dose: 1.5–15 kJ m^−2^	Increased total phenolics (25%). Hydroxycinnamoyl acid derivates were immediately increased after the treatments. The higher the UVC doses, the higher total antioxidant capacity values. UVC delayed chlorophyll degradation.	[106]
	Fresh-cut tatsoi baby leaves	Dose: 4.54 kJ m^−2^ with hyperoxia (100 kPa O_2_)	Improved phenolic content and total antioxidant capacity retention throughout storage. UVC and the combined UVC + O_2_- controlled the epiphytic microbes.	[107]
	Fresh-cut pomegranate arils	Dose: 4.54 kJ m^−2^	Combination of UVC and high O_2_ preserved SOD and CAT and decreased POD and PPO.UVC combined with high O_2_ maintained the level of anthocyanins and phenolics.Combining UVC to high O_2_ enhanced the benefits of applying each treatment alone.All treatments involving high O_2_ and/or UVC kept anthocyanins high, especially phenolic content.	[108]
		Dose: 4.54 kJ m^−2^	The lowest antioxidant activity was found in hot water + UVC + superatmospheric O_2_ packaging (HO) and the highest in UVC + HO and HO treatments. Hot water alone or in combination with UVC and HO inhibited mesophile, mold and yeast growth, while UVC + HO was most effective for controlling yeast and mold growth.	[109]
	Fresh-cut carrot	Intensity: UVB 9.27 W m^−2^, UVC 25.21 W m^−2^, Dose: UVB 1.5 kJ m^−2^. UVC 4.0 kJ m^−2^. Treatments alone or in combination	Combined UVC + UVB showed better results than each treatment alone.	[96]
	Fresh-cut red pepper	1.5; 3; 5; 6; 10 and 20 kJ m^−2^ in the inner (I), outer (O) or both fruit surfaces (I + O).	10 kJ m^−2^ (I + O) reduced decay and softening.UVC induced the accumulation of hydroxycinnamic acid-derivatives.Pectin solubilization and wall disassembly were delayed under UVC.UVC may control soft rots by modulating the host susceptibility.	[110]
	Garlic	Dose: 2.0 kJ m^−2^	Increased total phenolics (11%) and reduced microbial loads.	[111]
	Grape	Dose: 0, 0.5, 1.0, 2.0, or 4.0 kJ m^−2^	Increased activity of antioxidant enzymes (SOD and CAT) and induction of glutathione reductase and guaiacol peroxidase at longer times. Increased total thiol content by more than 2.0 kJ m^−2^, total phenolics (20%), anthocyanin (35%) for 5d at 20 °C.	[112]
	Red pepper	Dose: 10 kJ m^−2^	UVC treatments do not cause marked modifications in DPPH radical scavenging capacity or AA content. UVC treatments increase the activity of enzymes involved in the detoxification of superoxide and hydrogen peroxide (SOD, CAT and APX) during early cold storage.	[64]
		Dose: 6 kJ m^−2^ UV (B or C) and 6 + 6 kJ m^−2^ UV (B + C)	UVC greatly enhanced the flavonoid accumulation. UVC + UVB increased by ∼94% the carotenoid content and the flavonoid biosynthesis. Rutin accumulation was highly enhanced (∼70%).	[92]
	Spinach	Dose: 1.5–3 kJ m^−2^	Greatest induction of antioxidants (60%) and total phenolics (50%) with 1.5 kJ m^−2^	[113]
		Dose: 4.54–11.35 kJ m^−2^	Total antioxidant activity and polyphenols decreased throughout storage; this was more evident in higher UVC doses.Mesophilic and psychrophilic counts were reduced at similar level than conventional sanitization washing.	[114]
	Strawberry	Dose: 4.1 kJ m^−2^	Induction of anthocyanin biosynthesis and related enzymes, PAL, tyrosine ammonia-lyase and cinnamate 4-hydroxylase.	[115]
	Sweet cherry	Dose: 4 kJ m^–2^ or Interactions of UVC with 2 regulated deficit irrigation (RDI)	UVC increased phenols (21–36%) after shelf-life in RDI fruit.	[116]
		Dose: 1.0–4.2 kJ m^−2^	Induction of total phenolics, flavonoids, and anthocyanins (26%, 35% and 76% respectively). Induction of phenylpropanoid genes (PAL, C4H, 4CL).	[117]
	Tomato	Dose: 3.7 kJ m^−2^	Increased the accumulation of phenolic compounds and lignin.	[60]

**Table 2 foods-11-00653-t002:** Advantages and drawbacks of using UV radiation in foods.

Advantages	Drawbacks
Simple. Non-ionizing treatment. Approved by food control agencies. Strongly germicide (UVC) and broad microbiological control. Able to elicit hormetic responses inducing phytochemical accumulation in metabolically active foods. Relatively small changes in physicochemical quality attributes. Energy-efficient and cost-effective. Lower restriction than other irradiation methods. Could be combined with other preservation methods. No wastes or by-products generated. Does not require water.	Low penetration power in solids or turbid liquids. Little or no residual effect. Direct exposure required for germicide action and maximum effects. Absorbed by commonly used polymeric packing materials. Difficult to adapt to commercial operations/continuous processing. Harmful to operators if not properly protected. Consumers concerns although it is a non-ionizing radiation.

**Table 3 foods-11-00653-t003:** Factors affecting the efficacy of postharvest UV treatments in fruit and vegetables.

Product Variables	Process Variables
Commodity type	Radiation wavelength
Cultivar	Radiation dose (fluence)
Ripening stage	Radiation intensity (fluence rate/irradiance)
Degree of processing	Exposure pattern
Product–radiation interphase	Radiation uniformity
Product–microorganism interphase	Post irradiation illumination

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
