# Peer review of "Postharvest Ultraviolet Radiation in Fruit and Vegetables: Applications and Factors Modulating Its Efficacy on Bioactive Compounds and Microbial Growth"

_foods, 2022, doi:10.3390/foods11050653_

Round 1

Reviewer 1 Report

The objective of the review article is to demonstrate the main properties of UV technology for applications in the food sector, giving emphasis on post harvest application and effect on phytochemical elicitation. The study is interesting and of significant interest. The topic is focused enough so it can justify the need for a review article on an established technology, such as UV, on food products. This manuscript could also provide appropriate guidelines to the industry.

However, the authors should elaborate more on the structure of the manuscript and the points stated below, so as the manuscript can be acceptable for publication.

The title is too long. Please shorten the title and avoid using three sentences.

Section 1.2 should be placed later in the manuscript. The mode of action and potential mechanisms of UV towards the evaluated parameters should come first, this would facilitate the reading and understanding.

Figure 2 is not clear and does not provide adequate information to the reader. Please delete it or replace with a more representative scheme. Same comment for Figure 3.

The title of the sections should be revised. They are not adequate for a scientific article when stated as a question.

The authors should elaborate more on the writing of the review article, by providing critical comments on the available technologies and application of fresh produce. Potential limitations and drawbacks should be extensively discussed.

Author Response

Reviewer #1

 The objective of the review article is to demonstrate the main properties of UV technology for applications in the food sector, giving emphasis on post harvest application and effect on phytochemical elicitation. The study is interesting and of significant interest. The topic is focused enough so it can justify the need for a review article on an established technology, such as UV, on food products. This manuscript could also provide appropriate guidelines to the industry.

  • We thank the referee for the positive feedback regarding the article and for the valuable suggestions.

The title is too long. Please shorten the title and avoid using three sentences.

  • The title was shortened and modified considering the suggestions made. It now reads as “Postharvest Ultraviolet Radiation in Horticultural products: Applications, effects on Bioactive Compounds and factors modulating its efficacy”.

Section 1.2 should be placed later in the manuscript. The mode of action and potential mechanisms of UV towards the evaluated parameters should come first, this would facilitate the reading and understanding.

  • We have placed section 1.2 later in the manuscript as requested by the referee.

Figure 2 is not clear and does not provide adequate information to the reader. Please delete it or replace with a more representative scheme. Same comment for Figure 3.

  • The figures summarize the main uses of UV radiation in the food industry in general and in fresh produce. They are schematic and graphically summarize information that may be meaningful for some readers.

The title of the sections should be revised. They are not adequate for a scientific article when stated as a question.

  • We have modified the section titles as requested by the referee. Thank you.

The authors should elaborate more on the writing of the review article, by providing critical comments on the available technologies and application of fresh produce. Potential limitations and drawbacks should be extensively discussed.

  • Considering the feedback of all three reviewers we have decided not to expand further the article. We decided to focus the article on bioactive compounds to fit better with the topic of the special issue.  However, we have rearranged the manuscript and made some modifications to improve clarity. Thank you

Reviewer 2 Report

The subject of this paper, according to authors, was to discuss the most important properties of UV technology in relation to food applications, with emphasis on phytochemicals elicitation and retardation of microbial growth/senescence of fresh produce. The main mechanisms related to UV mode of action were also highlighted.

The review article is well written and gives a brief, but efficient overview of UV main advantages/disadvantages, as well as its principal applications in recent fruit and vegetable preservation. The structure is well laid out and the abstract reflects the key elements of the article content. Additionally, I believe that the manuscript has the novelty and the originality required for the specific scientific Journal.

Nevertheless, there are some points that need further analysis and some minor improvements are necessary for accepting the manuscript for publication.

To be more specific (and helpful), I would like to present in detail my points, and highlight the points to correct to authors:

 Regarding the title, I would suggest a different structure, to avoid using two sentences. For example, ‘Postharvest Ultraviolet Radiation: Applications, main effects on Bioactive Compounds and Microbial Growth and process factors modulating its efficacy’. Another point that has not been discussed, and deserves an analysis, is the use of UV technology as part of hurdle technology concept on fresh produce preservation. I would suggest to add a paragraph reporting some of the examples of combined application of this ‘hurdle’ as a part of a multiple procedure for extending fresh produce shelf life.

Line 15-16: try to avoid the repetition of the word ‘induce’

Line 19: what do you mean by the word ‘hormetic’?

Line 27: replace ‘extend’ by ‘extent’

Line 45: has been done..

Line 47: use a full stop, instead if a comma

Line 45-48: Check the sentence, something seems wrong with the syntax

  • 1.4 and 1.5: this part, which refers to some important UV applications need to be further analyzed…maybe provide a summarizing Table (similar to Table 2) describing the food tissue under study, UV media/process conditions and the main results obtained.

Line 178: the impact of postharvest..

Lines 216-218: Rephrase

Line 246: Use the appropriate numbering mode for references (instead of Jiang et al., 2012).

Lines 213-..: This paragraph reports really interesting findings on the role of factors of UV technology on bioactive compounds. Table 2 and its particular structure is very useful and contains the most important information. I would suggest to follow a similar approach for microbial growth (since these two applications, based on the title chosen, seem to be the most important according to authors). Maybe a similar Table for the application on the retardation of ripening and senescence.

Line 272: a closing bracket to references is missing

Lines 279-280: the sentence seems incomplete. Check and correct.

Line 291: have been reported..

Lines 299-303: Why do you use a red font?

In the Conclusion part, I believe that lines 371-386 are an unnecessary repetition of the Abstract.

Line  394: Despite the…, some important…

Line 399: Do you mean ‘tackling with…\? Rephrase this sentence, the meaning is not clear.

Author Response

-The subject of this paper, according to authors, was to discuss the most important properties of UV technology in relation to food applications, with emphasis on phytochemicals elicitation and retardation of microbial growth/senescence of fresh produce. The main mechanisms related to UV mode of action were also highlighted. The review article is well written and gives a brief, but efficient overview of UV main advantages/disadvantages, as well as its principal applications in recent fruit and vegetable preservation. The structure is well laid out and the abstract reflects the key elements of the article content. Additionally, I believe that the manuscript has the novelty and the originality required for the specific scientific Journal.

  • We greatly thank the reviewer for the positive comments regarding the manuscript.

 -Nevertheless, there are some points that need further analysis, and some minor improvements are necessary for accepting the manuscript for publication. To be more specific (and helpful), I would like to present in detail my points, and highlight the points to correct to authors:

 Regarding the title, I would suggest a different structure, to avoid using two sentences. For example, ‘Postharvest Ultraviolet Radiation: Applications, main effects on Bioactive Compounds and Microbial Growth and process factors modulating its efficacy’. Another point that has not been discussed, and deserves an analysis, is the use of UV technology as part of hurdle technology concept on fresh produce preservation. I would suggest to add a paragraph reporting some of the examples of combined application of this ‘hurdle’ as a part of a multiple procedure for extending fresh produce shelf life.

  • The title has been modified as recommended by Reviewers #1 and #2. We have also added a paragraph indicating that UV could be considered a useful tool in combination with other technologies as a part of the hurdle approach for food safety and preservation referencing a review on the topic (Esua et al 2020). The text now reads as: “UV irradiation is considered a very valuable tool within the hurdle technology, an integrated approach aimed in creating safe and stable foods by combining multiple physical, chemical, and/or biological preservation methods”.

 -Line 15-16: try to avoid the repetition of the word ‘induce’

  • The correction was made.

-Line 19: what do you mean by the word ‘hormetic’?

  • The word was changed by “beneficial”.

-Line 27: replace ‘extend’ by ‘extent’

  • The text has been corrected.

-Line 45: has been done..

  • The text has been corrected.

-Line 47: use a full stop, instead if a comma

  • A full stop has been added.

-Line 45-48: Check the sentence, something seems wrong with the syntax: 1.4 and 1.5: this part, which refers to some important UV applications need to be further analyzed…maybe provide a summarizing Table (similar to Table 2) describing the food tissue under study, UV media/process conditions and the main results obtained.

  • The sentence in lines 45-48 has been rewritten it now reads as: “Based on the known mechanisms of plant photoreception, the UVA region has been splitted into short-UVA (315-350 nm) and long-UVA (350-400) [8]”.

Regarding the suggestion of expanding sections 1.4 and 1.5 to further analyze on a product and process basis UV applications agree it would be useful. However, we believe such analysis would likely deserve a specific review article. Instead, in the present work considering the topic of the Special Issue we decided to focus the article on the effects of UVA, B and C radiation on  bioactive compounds rather than general and fully comprehensive review article.

-Line 178: the impact of postharvest..

  •  

-Lines 216-218: Rephrase

  • The sentence was splitted and rephrased. It now reads as: “Early in 1977 Langcake and Pryce [75] showed that UV exposure induced bioactive compounds in grape. This raised the interest in using UV treatments to improve fruit and vegetable phytochemical profiles”.

-Line 246: Use the appropriate numbering mode for references (instead of Jiang et al., 2012).

  • The reference was eliminated.

-Lines 213-: This paragraph reports really interesting findings on the role of factors of UV technology on bioactive compounds. Table 2 and its particular structure is very useful and contains the most important information. I would suggest to follow a similar approach for microbial growth (since these two applications, based on the title chosen, seem to be the most important according to authors). Maybe a similar Table for the application on the retardation of ripening and senescence.

  • We agree it would be useful to have tables reviewing the literature for most effects induced by UV radiation in fresh produce. Other review articles have already taken such approach. Then, as indicated we oriented the Review article to bioactive compounds to fit better the Special Issue.

-Line 272: a closing bracket to references is missing

  • The missing bracket was added.

-Lines 279-280: the sentence seems incomplete. Check and correct.

  • The sentence was checked as requested.

-Line 291: have been reported.

  • The text has been corrected.

-Lines 299-303: Why do you use a red font?

  • The font color has been changed.

-In the Conclusion part, I believe that lines 371-386 are an unnecessary repetition of the Abstract.

  • The conclusion section was shortened avoiding repetition with the abstract.

-Line  394: Despite the…, some important…

  • The text was corrected.

-Line 399: Do you mean ‘tackling with…\? Rephrase this sentence, the meaning is not clear.

  • The sentence was rewritten for the sake of clarity as requested by the referee.

Reviewer 3 Report

The review was directed towards the application of post-harvest UV treatments to enhance safety, shelf-life or nutraceutical levels. The area is relevant to the fresh produce sector and article relatively well written. There could have been a section on reactor design as this has been a more recent research focus. The authors could also consider pre-harvest UV treatments as this is an additional emerging area of interest.

Specific points

Title should be re-written. Assume it relates to fruit and vegetables but unclear.

The Abstract reads as a prolog to the introduction rather that providing the key conclusions from the review. This should be considered in the revised script.

Keywords: fruit, vegetables, food safety

Line 46: The lower wavelengths typically have less penetration power.

Line 58: Obviously preventing UV being exposed to those working in the area needs to be considered. In terms of advantages, UV has historically being applied for a wide range of applications and validation data is available.

Line 83: It is also possible to generate photoproducts that effect the quality and sensory attributes of some foods.

Fig 2: There are also UV-assisted processes. For example, UV-A with titanium dioxide, UV-assisted washing etc.

Line 124: Aseptic packing sterilization uses a combination of UV and hydrogen peroxide.

Line 126: Can also include UV pasteurization of juices.

Line 134: What doses would be needed to inactivate SARS-CoV-2?

Line 138: Not TiO2 as such but Advanced Oxidation Process.

Line 147: Also has pre-harvest applications.

Line 185: Should note that shading on surfaces limits the efficacy of UV light treatment of fresh produce. The other point being that the fresh produce sector still have a high level of dependence of the wash process.

Line 256: Speculative and would need to cite a reference to support the statement.

Table 3: Could also include the history of the fresh produce along with respiration rate and post-harvest storage conditions. For process variables, can consider duration and frequency of exposure, in addition to post-treatment dark hold.

Line 349: What do the authors consider low doses?

Line 365: Not entirely the case, there are UV tunnels and some interesting reactors based on LEDs to overcome shading. UV-assisted washing and/or hydrogen peroxide vapor has also been tested.

Author Response

Reviewer #3

-The review was directed towards the application of post-harvest UV treatments to enhance safety, shelf-life or nutraceutical levels. The area is relevant to the fresh produce sector and article relatively well written. There could have been a section on reactor design as this has been a more recent research focus. The authors could also consider pre-harvest UV treatments as this is an additional emerging area of interest.

Specific points Title should be re-written. Assume it relates to fruit and vegetables but unclear.

  • We agree with the referee that extra topics such as potential of preharvest UV or reactor design could have been interesting. However, we wanted to focus the article on the effects on bioactive compounds to fit better with the Special Issue.

 -The Abstract reads as a prolog to the introduction rather that providing the key conclusions from the review. This should be considered in the revised script.

  • The abstract was modified to include the main conclusions from the review as recommended.

-Keywords: fruit, vegetables, food safety

  • Food safety has been included as a keyword. The terms fruit and vegetables have not since they are now already present in the title based on reviewers #1 and 2 suggestions.

-Line 46: The lower wavelengths typically have less penetration power.

  • The reviewer is correct. We have included this point in the corrected manuscript.

-Line 58: Obviously preventing UV being exposed to those working in the area needs to be considered. In terms of advantages, UV has historically being applied for a wide range of applications and validation data is available.

  • We have added the point made by the referee into the text.

-Line 83: It is also possible to generate photoproducts that effect the quality and sensory attributes of some foods.

  • That is correct mostly for some type of foods. However, so far there are very few cases in which negative effects on sensory attributes have been modified in the case of fruit and vegetables, at least under the most common studied irradiation conditions.

-Fig 2: There are also UV-assisted processes. For example, UV-A with titanium dioxide, UV-assisted washing etc.

  • The referee is correct. We have pointed in the figure the most general applications for clarity.

-Line 124: Aseptic packing sterilization uses a combination of UV and hydrogen peroxide.

  • The reviewer is correct. Normally the cabinets are sterilized, and the actual packing materials are sterilized with H2O2.

-Line 126: Can also include UV pasteurization of juices.

  • We have changed the text to include UV juice pasteurization

-Line 134: What doses would be needed to inactivate SARS-CoV-2?

  • The conditions reported to inactivate SAR-CoV-2 have been determined in some studies but are still under investigation. We have added the reference of the work by Ma et al. (2021) which establishes that 1-0.3 kJ m-2 are required for 2 log cycle reductions depending on the region of the UV spectra used.

 Reference

Ma, B., Gundy, PM., Gerba, CP., Sobsey, MD., & Linden, KG. 2021. UV Inactivation of SARS-CoV-2 across the UVC Spectrum: KrCl* Excimer, Mercury-Vapor, and Light-Emitting-Diode (LED) Sources. Appl. Environm Microbiol. 87, e01532-21.

-Line 138: Not TiO2 as such but Advanced Oxidation Process.

  • The reviewer is right. We have corrected the sentence for the sake of clarity.

-Line 147: Also has pre-harvest applications.

  • The point has been included in the corrected manuscript.

-Line 185: Should note that shading on surfaces limits the efficacy of UV light treatment of fresh produce. The other point being that the fresh produce sector still have a high level of dependence of the wash process.

  • We have added the point made in the corrected manuscript.

 -Line 256: Speculative and would need to cite a reference to support the statement.

  • A reference reporting this statement has been added:

Reference

Katz, J., Yue, S., Xue, W. Increased risk for COVID-19 in patients with vitamin D deficiency. Nutrition 2021, 84, 111106. https://doi.org/10.1016/j.nut.2020.111106

 -Table 3: Could also include the history of the fresh produce along with respiration rate and post-harvest storage conditions. For process variables, can consider duration and frequency of exposure, in addition to post-treatment dark hold.

  • There is wide variation in the literature regarding the kind of measurements and information reported. Many studies do not have information regarding product history and respiration. In the present table to have consistency we selected most relevant operational conditions (dose, intensity, spectral region, time) as well as the product type. With regards to postharvest hold, we speculated that produce was held in the dark, but this is not specified in many published papers. Consequently, we believe it is more cautious and consistent to leave the Table in the present form.

-Line 349: What do the authors consider low doses?

  • Fresh produce has been UVB irradiated with doses ranging between 0,5 and ~ 300 kJ m-2 depending on the commodity treatment type (pre or postharvest, continuous or pulses) (Table 3). We consider low dosed below 5 kJ m-2. We have clarified this in the text.

-Line 365: Not entirely the case, there are UV tunnels and some interesting reactors based on LEDs to overcome shading. UV-assisted washing and/or hydrogen peroxide vapor has also been tested.

  • As far as we know these designs certainly interesting are still being developed and adjusted and not many commercial applications have reached the industry. However, it is an interesting point that has been added to the text.

Round 2

Reviewer 1 Report

The authors addressed adequately most of the reviewers' suggestions, however they did not considered the most important one. Review articles are in depth critical analysis of the state of the art from authors expert in the reviewed field, and the critical approach of the selected topic and proposed methodologies is a prerequisite for a high quality review article. the authors should elaborate more on this important aspect of review articles and provide a more in-depth critical analysis, as indicated by the reviewer at the 1st round of the review process.

Author Response

 We have taken into account most recommendations made and incorporated them into the text. With regard to expanding the text to make a more comprehensive discussion of all the points suggested we believe it would reduce the focus of the current article which is, in our view, better suited for a special issue on Bioactive compounds.